# Imaging and Image Processing Techniques for High-Resolution Visualization of Connective Tissue with MRI: Application to Fascia, Aponeurosis, and Tendon

**DOI:** 10.3390/jimaging11020043

**Published:** 2025-02-04

**Authors:** Meeghage Randika Perera, Graeme M. Bydder, Samantha J. Holdsworth, Geoffrey G. Handsfield

**Affiliations:** 1Auckland Bioengineering Institute, University of Auckland, Auckland 1010, New Zealand; mper162@aucklanduni.ac.nz; 2Mātai Medical Research Institute, Tairāwhiti-Gisborne 4010, New Zealand; gbydder@health.ucsd.edu (G.M.B.); s.holdsworth@matai.org.nz (S.J.H.); 3Department of Radiology, University of California, San Diego, CA 92093, USA; 4Department of Anatomy and Medical Imaging, Faculty of Medical and Health Sciences & Centre for Brain Research, University of Auckland, Auckland 1010, New Zealand; 5Department of Orthopaedics, School of Medicine, University of North Carolina at Chapel Hill, Chapel Hill, NC 27514, USA; 6Joint Department of Biomedical Engineering, University of North Carolina at Chapel Hill, Chapel Hill, NC 27514, USA; 7Joint Department of Biomedical Engineering, North Carolina State University, Raleigh, NC 27606, USA

**Keywords:** MRI, fascia, ultra-short-T_2_*, medical image processing

## Abstract

Recent interest in musculoskeletal connective tissues like tendons, aponeurosis, and deep fascia has led to a greater focus on in vivo medical imaging, particularly MRI. Given the rapid T_2_* decay of collagenous tissues, advanced ultra-short echo time (UTE) MRI sequences have proven useful in generating high-signal images of these tissues. To further these advances, we discuss the integration of UTE with Diffusion Tensor Imaging (DTI) and explore image processing techniques to enhance the localization, labeling, and modeling of connective tissues. These techniques are especially valuable for extracting features from thin tissues that may be difficult to distinguish. We present data from lower leg scans of 30 healthy subjects using a non-Cartesian MRI sequence to acquire axial 2D images to segment skeletal muscle and connective tissue. DTI helped differentiate aponeurosis from deep fascia by analyzing muscle fiber orientations. The dual echo imaging methods yielded high-resolution images of deep fascia, where in-plane spatial resolutions were between 0.3 × 0.3 mm to 0.5 × 0.5 mm with a slice thickness of 3–5 mm. Techniques such as K-Means clustering, FFT edge detection, and region-specific scaling were most effective in enhancing images of deep fascia, aponeurosis, and tendon to enable high-fidelity modeling of these tissues.

## 1. Introduction

Deep fascia, aponeurosis, and tendons are musculoskeletal connective tissues that are rich in collagenous proteins [1,2,3]. Because of their chemical structure, T_2_* decay is rapid, and conventional MRI protocols are not capable of acquiring signals in these tissues due to echo times that are not sufficiently short. Additionally, for deep fascia and certain regions of aponeurosis, high spatial resolution is essential since these tissues may be very thin. To generate useful images of these connective tissues, imaging methods must feature ultra-short TEs, generate contrast between connective tissues and their neighboring structures, and—for deep fascia and thin regions of aponeurosis—achieve submillimeter spatial resolution. Modern coil setups and multiple signal averages may maximize the signal-to-noise ratio. Image post-processing techniques may further improve image quality and emphasize specific features that are important for further analysis and modeling.

### 1.1. Image Acquisition—Ultra-Short Echo Time MR Imaging

Ultra-short TE is an imaging technique that overcomes the short T_2_* relaxation times of tissues by acquiring signals at TEs much shorter than 1 ms. Dual-echo UTE has been shown to produce high-signal and high-contrast images of collagen-rich tissues [4,5,6]. This technique generates two echoes for signal acquisition: one at an ultra-short time (e.g., 0.05 ms) and the second with a more conventional TE (e.g., 7.2 ms). The ultra-short TE captures signals from deep fascia and adjacent tissues, such as muscle, fat, and water, while the longer TE is optimized to capture signals from other musculoskeletal tissues, particularly muscle. The second TE is set sufficiently long to avoid signal acquisition from deep fascia or other collagenous connective tissues. By properly optimizing both echoes, image subtraction yields a final image with a high connective tissue signal and improved contrast relative to neighboring tissues. With optimal selection of imaging parameters and coil positioning, this approach produces images with high contrast and sufficient spatial resolution to identify deep fascia or thin regions of aponeurosis in the musculoskeletal system.

### 1.2. Diffusion Tensor Imaging (DTI)

Over the last two decades, the application of DTI to studying skeletal muscle fiber orientation has expanded [7,8,9]. DTI in muscles has been demonstrated in skeletal muscles spanning the calf [10,11,12,13,14], thigh [15,16,17], forearm [18,19], tongue [20,21], feet [22], ocular muscles [23], and small pelvic muscles [24,25,26]. DTI fiber tractography may be used to generate 3D models of muscle architecture, facilitating the measurement of parameters such as physiological cross-sectional area (PCSA), fiber length, and pennation angle [8,27,28,29].

From the perspective of connective tissue imaging, one tantalizing benefit of DTI is that fiber tract data produced from DTI can be used to distinguish between aponeurosis and deep fascia. These two tissues may appear similar with dual-echo UTE MRI based on their similar chemical properties [30]. Both tissues also lie on the periphery of skeletal muscles. Functionally, one difference that exists is that aponeurosis abuts the endpoints of muscle fascicles, which transmit force into the aponeurosis. Deep fascia, on the other hand, lies adjacent to skeletal muscle fascicles. Determining the location of muscle fascicle endpoints thus may help to distinguish deep fascia from aponeurosis, and DTI tractography of muscle may serve to make this distinction.

DTI processing involves a series of steps, including denoising, correcting for Gibbs ringing, addressing eddy current effects, bias correction, anatomical and structural image alignment, and ultimately generating streamlines. Tools within software such as MRtrix3 3.04 [31], FSL [32], and DSI Studio [33] offer the essential functionalities needed to achieve successful tractography.

### 1.3. Image Processing

As computer hardware systems have advanced, leveraging computer vision for medical research has become increasingly feasible. In this study, we utilized ITK-SNAP, an application built on the Insight Toolkit (ITK), to segment the deep fascia—a thin tissue layer located directly outside the muscle. The segmentation of this structure is particularly challenging due to its small size and low contrast in MR images. To address variations in magnetic field and tissue intensity, preprocessing steps such as low-pass filtering and histogram binning were applied to enhance image quality. Additionally, edge-detection techniques like FFT-based filtration were employed to improve segmentation accuracy by better delineating tissue boundaries. Our work also explored applying these segmentation techniques to aponeuroses, which face similar challenges in regions where the tissue is thin. After segmentation, we measured key structural features, including the thickness of the deep fascia, as well as lower leg muscle volume and cross-sectional areas. These measurements provide valuable insights into tissue structure and serve as the foundation for further statistical analysis. The workflow for this study involved the following steps:Acquiring images in DICOM.Converting image formats to ensure compatibility with analysis tools.Preprocessing images to ensure consistency and enhance contrast through bias correction, registration, and normalization using MATLAB.Performing manual segmentation of the deep fascia and aponeuroses in ITK-SNAP.Taking measurements from segmented images, including thickness of the deep fascia, lower limb muscle volume, and muscle cross-sectional areas.Statistical analysis of measurements.

This approach provides a framework to analyze connective tissue structure from magnetic resonance images.

This study employs both supervised and unsupervised segmentation approaches to enhance tissue contrast and ensure accurate deep fascia identification. The main aims of this study are to (1) propose and assess an image processing pipeline that simplifies connective tissue segmentation by improving the contrast of the tissues of interest; (2) perform fascia thickness measurements and compare to the available literature; and (3) explore the integration of ultra-short echo time (UTE) MRI sequences with Diffusion Tensor Imaging (DTI) for improved visualization and analysis of musculoskeletal connective tissues like tendons, aponeurosis, and fascia. The ultimate goal is to evaluate the reliability of these techniques in improving tissue visualization, thereby making manual and semi-automatic segmentation more efficient and less time-consuming.

## 2. Methods

### 2.1. Optimized Ultra-Short Echo Time MRI

The imaging and recruitment protocols described here were approved by The University of Auckland Human Participant Ethics Committee. Volunteers were scanned using one of two 3T MRI scanners at different sites: a 3T GE Signa Premier and a 3T Siemens Skyra system. Participants were positioned feet-first supine, with knees nearly fully extended in a comfortable position. The entire right leg was scanned from hip to ankle using five distinct stations. The GE system used a dual-echo sequence with a 3D radial k space trajectory, aka 3D Cones [34]. A 15-channel knee coil was used with the following sequence parameters: FOV = 180 mm × 180 mm, TE1(UTE)/TE2(shTE) = 0.028 ms/2.2 ms, TR = 70 ms, matrix size 256 × 256, and spatial resolution 0.7 mm × 0.7 mm × 4 mm. A second sequence was used on the GE system with the following parameters: FOV = 205 mm × 205 mm, TE1(UTE)/TE2(shTE): 0.028 ms/2.2 ms, TR = 60 ms, matrix size 256 × 256, and spatial resolution 0.8 mm × 0.8 mm × 4 mm. A modified version of 3D Cones was used to boost the resolution to 0.3 mm × 0.3 mm × 4 mm. This sequence is known for its zero TE (ultra-short echo time), which we used with a more-than-doubling of image resolution. The remaining sequence parameters are as follows: FOV = 186 mm × 186 mm, TE1(UTE)/TE2(shTE): 0.0032 ms/4.4 ms, TR = 70 ms and matrix size 620 × 620. On the 3T Siemens Skyra, a 2D stack of spirals sequence (Spiral VIBE [35]) was used with parameters FOV = 320 mm × 320 mm, TE1(UTE)/TE2(shTE): 0.05 ms/5.19 ms, TR = 15 ms, matrix size 640 × 640, and spatial resolution 0.5 mm × 0.5 mm × 5 mm.

### 2.2. Image Processing Pipeline

The images from the 3D Cones UTE sequence were further processed to enhance the contrast of deep fascia and generate composite images. We present several image processing pipelines designed to enhance image contrast and detect thin connective tissue, tailoring the approach to the quality of the dataset and the characteristics of the tissue. The following section explains different approaches that have been successfully employed so far.

#### 2.2.1. Pipeline of Operations Used in This Section

The following methods were combined into a singular workflow for fascia imaging and image processing (Figure 1). The image processing pipeline is divided into two main phases: the image enhancement/noise removal phase and the image segmentation phase. The segmentation phase involved three primary methods, all of which required initial image enhancement and noise removal. Following this, watershed and FFT methods were applied. K-Means clustering was implemented after FFT for improved results. FFT was primarily used to enhance tissue boundaries, while K-Means clustering effectively distinguished most connective tissue layers from neighboring tissues. The watershed method, on the other hand, was particularly effective in isolating the deep fascia layer, which varies in thickness across the geometry. The goal of this approach was to utilize these techniques as tools for assisting manual segmentation—rather than relying on them as standalone solutions for deep fascia segmentation—to minimize segmentation errors. However, for some datasets, automatic segmentation was feasible for most slices, depending heavily on the acquisition quality.

#### 2.2.2. Region-Specific Scaling

In UTE imaging, there is a lower signal-to-noise ratio than in imaging with longer TEs within the short TE category [30]. However, by matching signal intensities of non-target tissues between the ultra-short TE images and the short TE images, subtraction may then be optimized to remove signal from the surrounding muscle and highlight the layer of deep fascia, thus increasing contrast. To achieve this, all image pixel values needed to be increased by some factor. This factor may be determined by comparing the computed mean and median of the images. This process is presented in the form of a flowchart in Figure 2.

#### 2.2.3. Other Preprocessing Techniques

Noise filtration is an essential step during image preprocessing methods [36]. Improving the signal-to-noise ratio (SNR) augments the ability to accurately segment tissues of interest. In contrast to erosion and dilation operations, which can alter thickness measurements of deep fascia and aponeurosis, the Wiener filter is a preferred alternative that is effective when dealing with noisy signals [37].

The specifics of the noise removal and image enhancement techniques, along with their settings, are provided below.

The average and standard deviation of intensity for each input image are computed as initial statistical parameters. Contrast enhancement is performed using the imadjust function, leveraging these parameters to optimize the image’s intensity distribution. Subsequently, a Top Hat morphological operation is applied to the contrast-enhanced image using a disk-shaped structuring element with a radius of 10 to emphasize smaller, brighter features. Noise reduction is then conducted using the Wiener filter (wiener2) with a window size of 20 × 20.

#### 2.2.4. Edge Detection Method Using FFT

Perhaps the most useful image processing method for deep fascia image processing in this work is edge detection. Since deep fascia is located at the boundary of the muscle, it is important to extract edges in order to enhance the features of the image. Edge detection is a useful algorithm prior to image segmentation. Feature extraction is conducted in the frequency domain, where sharp edges appear as high-frequency components. After applying a suitable FFT operation, low-pass filtering is used to remove unnecessary components of the image.

Once the grayscale image is transformed into the frequency domain, the low-frequency components are centered for better visualization and manipulation. The log-magnitude of the frequency spectrum is computed to enhance the visibility of noise elements, which appear as bright spikes. A binary mask is applied to isolate high-amplitude noise components, while the central region containing essential low-frequency information is preserved. The identified noise spikes are suppressed to reduce periodic noise, and the filtered frequency spectrum is transformed back into the spatial domain to reconstruct the denoised image. Finally, the image is normalized to a standard range and converted to a format suitable for further analysis.

#### 2.2.5. K-Means Clustering

K-Means clustering identifies regions based on intensity similarities, grouping pixels with similar values. This method efficiently segments regions that are difficult to resolve with the naked eye and is suitable for identifying thin layers of deep fascia. The quality of the output depends on the selection of the proper number of clusters and centroid locations. These parameters were tuned by trial and error to improve segmentation. Below are the steps involved in the K-Means clustering algorithm:Randomly select k data points as initial centroids.Assign each data point to the nearest centroid based on the Euclidean distance between the two points.Update the centroids based on the mean of all data points assigned to each centroid.Check for convergence: if the centroids have changed significantly, continue updating; if not, stop the iteration.Finalize centroid values.

Prior work in musculoskeletal research has used K-Means clustering to quantify adipose tissue and muscle volumes in the lower limbs [38,39]. In our study, we expanded the workflow to include the identification of an extra layer, namely the deep fascia layer. The trial-and-error process is a crucial step for fine-tuning parameters; therefore, the settings used in the pipeline are shared here. The kmeans function requires data to be in a one-dimensional format, so the original data were reshaped accordingly. The number of clusters selected was 3. To ensure stability, the clustering process was repeated 10 times (replicates = 10), minimizing the within-cluster sum of squares. The output cluster indices (idx), which assign each pixel to a specific cluster, were reshaped back to their original dimensions. The final clustered image was visualized using a color-coded representation that groups pixels with similar intensities, offering a simplified representation of the image for segmentation or analysis.

In this workflow, the process begins with the use of the FFT method for edge detection, as outlined in Section 2.2.4. This technique highlights significant intensity changes in the image while helping to eliminate unnecessary noise, resulting in a cleaner signal. The denoised image then serves as a more effective input for K-Means clustering, allowing it to more accurately group pixels with different intensity levels. These clusters primarily represent the fascia, muscle, and fat layers, thereby improving the accuracy and relevance of the tissue segmentation.

#### 2.2.6. Watershed Method

The watershed method is effective in detecting complex structures with irregular shapes within images [40]. Deep fascia is not evenly distributed around the periphery of muscle tissue. This technique has advantages in directly identifying these differences.

The watershed algorithm represents image gradients as elevation variation, e.g., in a landscape with low points corresponding to low-intensity differences of gray level across the image, while high points represent significant intensity differences [41]. The watershed algorithm is used to segment image data into regions based on markers provided by the user. In our case, it segments the muscle, deep fascia, and fat layers. This method is an oft-used muscle segmentation technique in the literature [42,43].

Initially, the grayscale image was binarized using a threshold of 0.6 (im2bw (I4, 0.6)), creating a binary mask BW where pixels above the threshold were marked as foreground (1) and others as background (0). The Euclidean distance from each foreground pixel to the nearest background pixel was calculated using the distance transform (bwdist). This distance map was negated (−bwdist(~BW)) to identify basins for the watershed algorithm. Non-foreground regions were set to -Inf to prevent the algorithm from segmenting areas outside the foreground. The watershed transformation was then applied to the modified distance map (watershed(D)), which segmented the regions based on intensity ridges. Background areas were masked out by resetting non-foreground pixels to zero (L(~BW) = 0).

### 2.3. Image Segmentation and Fascia Thickness Calculation

Image segmentation was undertaken using a personal computer (MacBook Pro; Apple, Cupertino, CA, USA). T1-weighted images were used for axial plane segmentation using ITK-SNAP [44] (http://www.itksnap.org/pmwiki/pmwiki.php—accessed on 29 January 2025). We used MATLAB for data processing, leveraging its built-in filter and image processing functions. The primary effort involved the author’s expertise in fine-tuning the parameters of these filters and image processing algorithms to achieve the clearest possible segmentation with minimal noise. Additionally, an in-house MATLAB algorithm, as described in a recent paper [45] which inspired by [46], was applied to the segmentation mask to calculate the thickness of the fascia. A brief explanation of the algorithm is provided here: The program determines the skeletal structure of the tissue, representing the midline, and measures the distance from this line to the inner and outer tissue boundaries at multiple points. These distances are averaged to calculate the thickness for each slice. The process is repeated for all slices, and the overall fascia thickness is obtained by averaging the measurements across slices.

### 2.4. Diffusion Tensor Imaging

Both 3D-T1-weighted gradient echo sequence and DTI sequences were performed with the identical scanning setup used for UTE scanning. The subjects were positioned in a comfortable, supine, feet-first position with their knee joints nearly fully extended.

The following parameters were used for T1-weighted imaging: TE = 38 ms, TR = 4780 ms, slice thickness 5 mm, matrix size = 1024 × 1024, FOV = 250 × 250 mm, in-plane resolution = 0.5 mm × 0.5 mm. For DTI, a spin echo echoplanar imaging (EPI) sequence was used with slice thickness = 5 mm, in-plane resolution 1.3 × 1.3 mm, 64 slices, TR/TE = 2887/49 ms, number of signal averages = 1, diffusion directions = 20, b = 500 s/mm^2^, and a reference image with b = 0 s/mm^2^.

Segmented masks were used to create ROIs for DTI analysis; these coincided with the muscles of interest. Preprocessing of DTI data involved denoising and correcting for eddy current distortions, B0-inhomogeneity, and rigid motion correction using algorithms built into MRTrix [31,46]. A slight discrepancy between T1-weighted and DTI images motivated a registration procedure between the two datasets using FSL’s FLIRT tool [47]. The resulting transformation matrix was applied to the segmentation masks to fit them into the same image space.

#### DTI Tractography

FA and MD maps were calculated at each voxel using MRtrix (version 3; Brain Research Institute, Melbourne, Australia; (https://www.mrtrix.org/—accessed on 29 January 2025). [47]. Registered segmented masks (see previous section) were used, and tractography was undertaken. Tracking parameters were adopted from previous work [48] to decide on the start and stop conditions for the tractography process. The algorithm generated 1000 fiber tracts with lengths ranging from 5 to 200 mm. The binarized mask was used as the seed image, and the process terminated once the generated fiber crossed any voxel outside the mask. DTI data were also used to compute muscle architecture parameters of fascicle length, pennation angle, and muscle physiological cross-sectional area (PCSA). Fascicle length was defined here as the length between the origin of the fascicle and the insertion extrapolated by the algorithm [49]. The pennation angle is the angle between the muscle’s line of action vector and each extrapolated fascicle vector [50]. Muscle PCSA was calculated by dividing the muscle volume by the mean fascicle length [50].

## 3. Results

### 3.1. Ultra-Short TE Sequence Results

Results demonstrated high-resolution and high-contrast images of deep fascia in the lower limb (Figure 3), as well as in the aponeurosis and tendon (Figure 4). The locations of peripheral high-signal tissue we observed in our images were assessed and confirmed as deep fascia by the radiologist on our study team (GMB).

### 3.2. Image Processing Method Results

Results from image processing methods reveal enhanced contrast for deep fascia tissue in the lower limb from region-specific scaling, intensity images, Weiner filters, edge detection using FFTs, watershed, and K means clustering (Figure 5, Figure 6 and Figure 7). In cases where deep fascia is not detectable due to insufficient contrast, these methods may be used to enhance contrast for deep fascia identification, segmentation, or geometrical measurement (e.g., thickness). We found that region-specific scaling improved SNR by 65% (Figure 7).

Inter-observer variability tests were conducted with four other trained researchers to evaluate segmentation precision. The fascia thickness segmentation yielded a Dice Coefficient of 0.904 and a Jaccard Index of 0.825, indicating good precision.

We assessed how these results align with one another and with gold standard dissection measurements [51,52,53,54,55]. Figure 8 illustrates how the various techniques we employed compare to the normal range of lower-leg deep fascia thickness. All the image processing methods applied fall within the normal deep fascia thickness range reported in the literature, which serves as the gold standard. However, the UTE data exhibited significantly higher thickness values due to the challenges in accurately segmenting it without implementing advanced image processing techniques.

### 3.3. Diffusion Tensor Tractography Results

Diffusion Tensor Imaging was used to determine muscle fiber directions within specific muscle regions, as determined by segmentation and mask generation. Tractography produced representations of skeletal muscle fiber directions consistent with anatomy for lower limb muscles, e.g., the medial gastrocnemius (Figure 9). This method revealed fascicle lengths of 59.87 ± 26.31 mm and a pennation angle of 22.37 ± 10.06° (Figure 9), which aligns with values reported in the literature [56,57].

## 4. Discussion

Tendons, aponeuroses, and the deep fascia surrounding skeletal muscles appear as dark regions in conventional MRI due to the rapid T_2_* decay of these tissues and limitations in acquiring ultra-short TEs. This paper presents images acquired using dual-echo UTE protocols on 3T MRI that reveal connective tissues in the human lower limb, including tendon, aponeuroses, and deep fascia. Previous work has shown the utility of dual-echo UTE for imaging tendons [34,58,59,60] and aponeuroses [61,62]. To our knowledge, this is the first demonstration of deep fascia imaging in humans using dual-echo UTE. A previous study used 3D UTE to achieve high signal intensity and contrast in visualizing the deep peripheral fascia in a sample of the porcine lower leg [30,63] and radiological findings in humans [62], which supports the utility of this approach. Additionally, a comparative study on Dixon imaging and UTE for aponeurosis identification found that UTE performed better [61], providing clearer visualization and enhanced contrast for accurate delineation. Dual-echo UTE—in the case of the present study with 3D Cones and Spiral VIBE sequences—holds utility for connective tissue imaging. With increases in image resolution, increasingly thinner regions of deep fascia may be imaged with fewer artefacts, which has exciting potential for deeper in vivo exploration of this tissue.

The work presented here involved three imaging protocols: a dual-echo UTE, a T1-weighted scan, and DTI (Figure 10). Tractography data from DTI may be used to provide context to connective tissue imaging—the endpoints derived from DTI tracts indicate regions corresponding to deep fascia versus regions corresponding to aponeuroses, where aponeurosis tissue coincides with regions of tract endpoints from DTI. Additionally, DTI data may be used to compute parameters of muscle architecture, such as fascicle length, pennation angle, and physiological cross-sectional areas.

The high-resolution deep fascia imaging that resulted from utilizing a dual-echo UTE MRI sequence contributes significantly to the way we may look at anatomic structures in the future. This is an example of the results that can be achieved from the advancement of imaging protocols, as we can now observe a tissue that was generally imperceptible using previous methods [5]. The ability to recognize deep fascia with high resolution not only enhances our ability to understand the architectural structures but opens many more avenues for clinical and research applications. Although several studies [63,64,65] have analyzed pathology associated with deep fascia, such as tumors and trauma, these often rely on fat-suppressed T1-weighted images. However, the studies lack the spatial resolution to correctly diagnose the pathologic changes within deep fascia. This could be another important advantage of this work, which has the potential to distinguish between healthy and compromised deep fascia. The incorporation of DTI into this study was principally to differentiate between deep fascia and aponeurosis. DTI is capable of capturing muscle fiber microstructure and fiber orientations [29,65], which could be used here to accurately differentiate neighboring structures.

This study, which is related to deep fascia imaging, has the potential to be applied in various clinical contexts, particularly for assessing pathological conditions. For example, these methods could aid in the diagnosis of deep fascia-related injuries, such as iliotibial band syndrome, myofascial pain syndrome, plantar fasciitis, and compartment syndromes, by enabling detailed imaging of deep fascia structures. Additionally, since these imaging techniques can also capture aponeurosis tissues, they may be useful in identifying aponeurosis tears, which are commonly observed in athletes. Although further research is necessary to confirm the effectiveness of these methods, these advancements hold promise for improving the diagnosis and management of such conditions.

There are several limitations of this study. In this study, we relied on recently published in vivo cadaver and in vivo pig studies to verify that our fascia thickness measurements were reasonable. While these studies provide valuable insights and benchmarks, they do not confirm that our thickness measurements from MRI are absolutely correct. Further cadaver studies may provide confidence in this method and measurements drawn from it. Additionally, pathological tendon, aponeurosis, and deep fascia tissues were not investigated. It would be a worthwhile future area of research to use UTE imaging to find suitable biomarkers to distinguish between healthy and unhealthy deep fascia. Scan times and hardware limitations for producing very short UTE pulses constrained our methods. Further optimization and hardware development may propel connective tissue imaging, particularly deep fascia imaging, including improvements in spatial resolution. Another limitation is the reliance on multiple software tools (MATLAB, ITK-SNAP, and MRtrix) to implement this pipeline, as each software offers specific functionalities contributing to this workflow. Future development of an automated, standalone pipeline could improve the ease of implementation and serve in the creation of biomechanical models. Additionally, the current workflow required some trial-and-error tuning of imaging and image processing parameters to achieve high-contrast results; readers are advised to approach future imaging and image processing of UTE data with the expectation that heuristic optimization may be needed. Future work incorporating machine learning approaches may help to automate parameter selection. Lastly, we applied image processing slice by slice in a 2D fashion rather than on the entire 3D dataset due to the anisotropy in voxel size. Future 3D image processing is worth exploring.

## 5. Conclusions

The images showcased in this study demonstrate ultra-short echo time MRI with sufficiently high in-plane resolution (0.5 × 0.5 mm) for imaging the musculoskeletal connective tissues, including tendons, aponeuroses, and deep fascia. Furthermore, T1-weighted and DTI MRI sequences were employed for segmentation, mask generation, and determination of muscle fiber directions to distinguish between deep fascia and aponeurosis. Advanced image processing techniques were employed to enhance the contrast of the connective tissues, followed by connective tissue segmentation and thickness measurements. These structural parameters are likely to be important for identifying biomarkers capable of distinguishing between healthy and compromised deep fascia, aponeuroses, and tendons.

## Figures and Tables

**Figure 1 jimaging-11-00043-f001:**
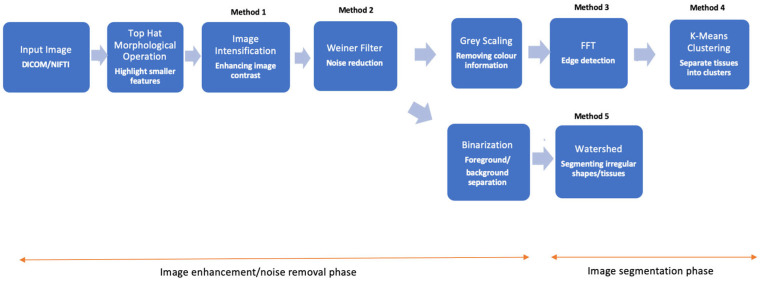
Flowchart depicting the image analysis workflow. The process starts with input images in DICOM/NIfTI format and proceeds through image enhancement steps such as Top Hat morphological operations, image contrast enhancement, and Wiener filtering for noise reduction. The segmentation phase applies different methods such as FFT for edge detection, binarization for foreground/background separation, watershed for irregular tissue segmentation, and K-means clustering for tissue classification.

**Figure 2 jimaging-11-00043-f002:**
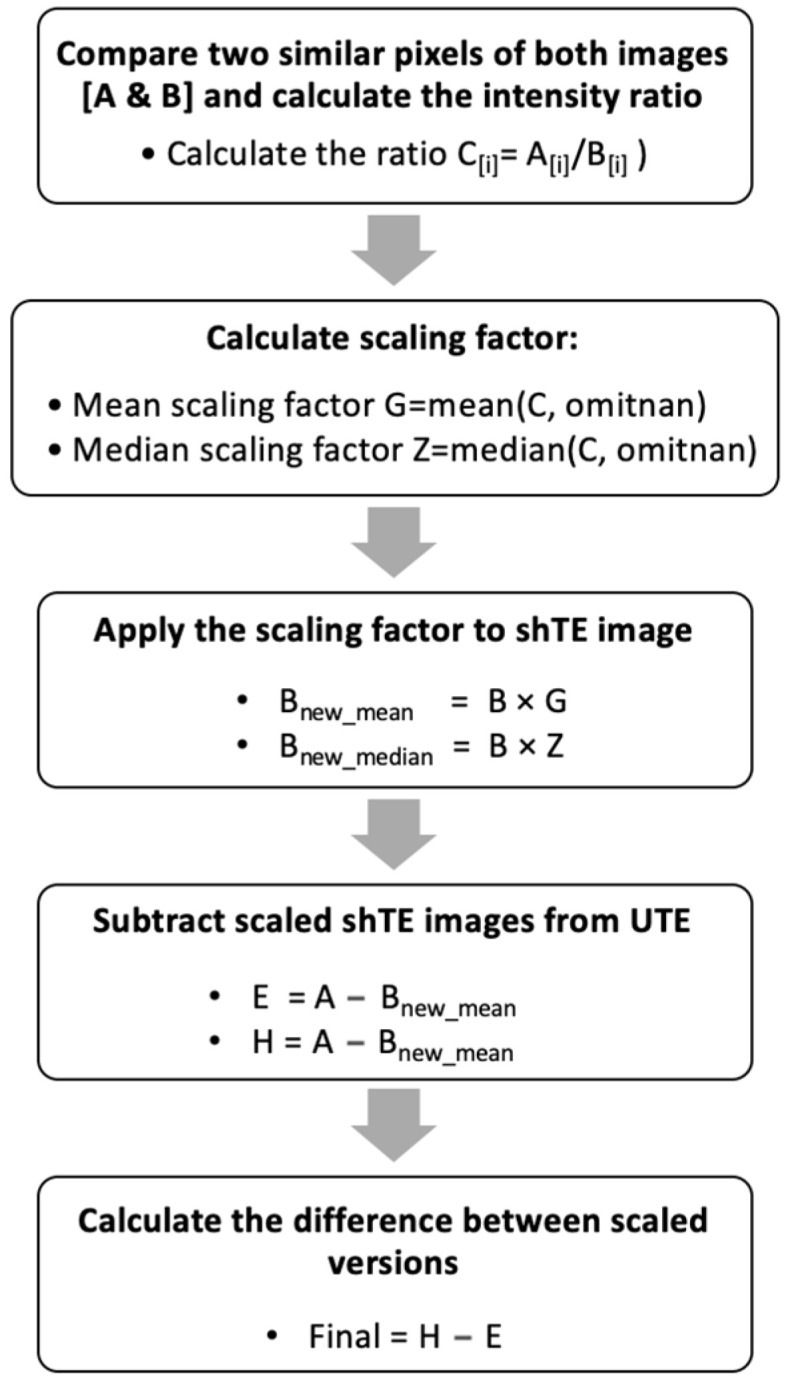
Workflow of contrast difference reduction method.

**Figure 3 jimaging-11-00043-f003:**
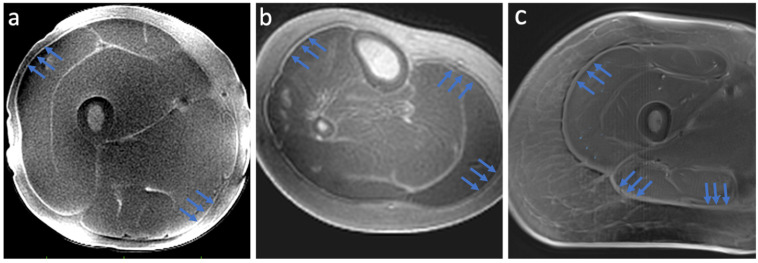
Results from the dual-echo MRI sequences designed for deep fascia imaging in the lower limb. The 3D Cones sequence in the (**a**) upper leg and (**b**) lower leg. The Spiral VIBE sequence produced similar images showing deep fascia (**c**). Blue arrows indicate deep fascia tissue.

**Figure 4 jimaging-11-00043-f004:**
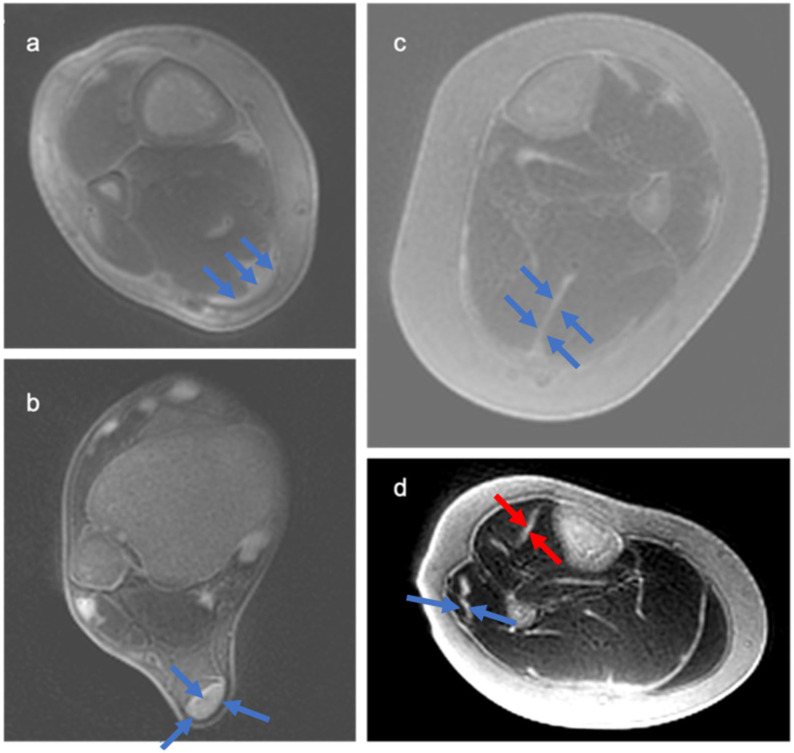
Examples of tendon and aponeurosis imaging using dual-echo UTE sequences. (**a**) Axial images from the 3D Cones sequence show the Achilles tendon aponeurosis (blue arrows), which is proximal to the free tendon. (**b**) Free Achilles tendon is seen (blue arrows) in an image acquired using the Spiral VIBE sequence. (**c**) The median septum of the soleus—part of the Achilles aponeurosis—is shown (blue arrows) in axial images acquired using the 3D Cones sequence. (**d**) Aponeurosis of the peroneus longus muscle (blue arrow) and tibialis anterior (red arrow) is shown; images acquired with 3D Cones sequence.

**Figure 5 jimaging-11-00043-f005:**
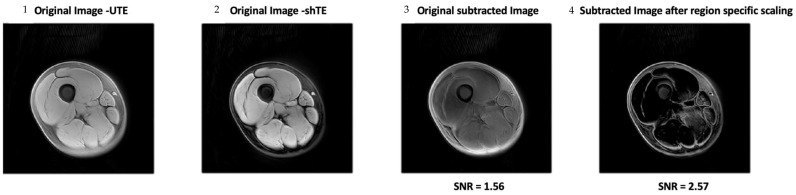
A comparison of MRI images demonstrating the effects of a region-specific scaling technique. The sequence includes (1) original image—UTE, (2) original image—shTE, (3) subtracted image with SNR = 1.56, and (4) subtracted image after applying region-specific scaling with SNR = 2.57. The significant improvement in SNR highlights the effectiveness of this technique in enhancing deep fascia contrast.

**Figure 6 jimaging-11-00043-f006:**
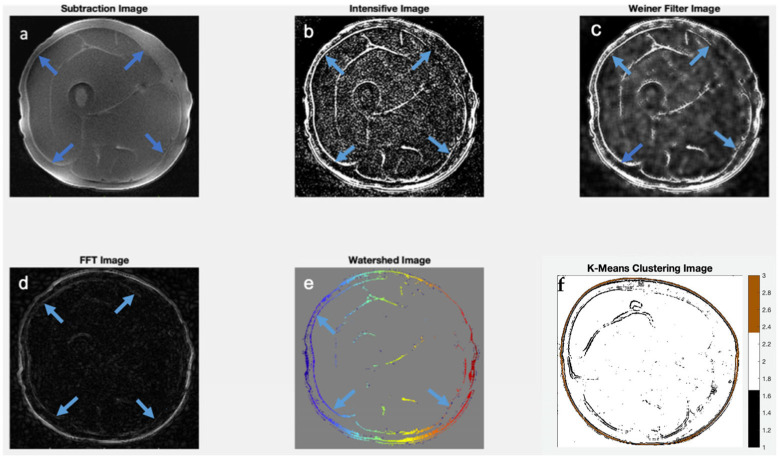
Image processing techniques demonstrate enhanced contrast for deep fascia identification. Blue markers on the images show deep fascia locations. (**a**) Subtraction of the short echo time image from the ultra-short echo time image reveals deep fascia on the periphery of the muscle using a 3D Cones sequence. (**b**) Region-specific scaling on image from (**a**) enhances deep fascia–muscle contrast. (**c**) Application of the Weiner filter improves contrast. In subfigures (**d**–**f**), the ROI—in this case, deep fascia—was extracted from the images. (**d**) Edge detection algorithm based on FFT on preprocessed 3D spiral VIBE image filters out deep fascia within muscle. (**e**) Watershed algorithm used to segment the features using unsupervised learning; in this case, the deep fascial layers. (**f**) K-Means clustering highlights deep fascia regions as well (shown here in the black channel).

**Figure 7 jimaging-11-00043-f007:**
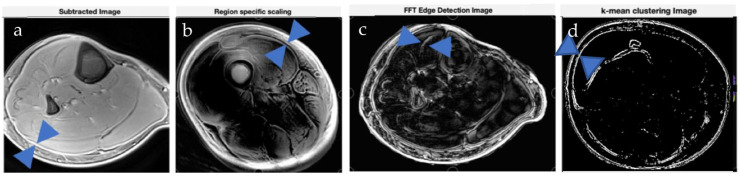
Illustration of image processing methods applied to various connective tissues. Blue markers indicate the locations of the deep fascia and aponeurosis, which are manually annotated to help the reader easily identify the relevant tissues. (**a**) Image subtraction reveals high-signal, high-contrast images of deep fascia abutting the periphery of the muscle. (**b**) Region-specific scaling enhances deep fascia–muscle contrast. (**c**) Edge detection algorithm on 3D spatial image reveals aponeurosis within muscle. (**d**) K-Means clustering highlights the deep fascia region, shown in blue.

**Figure 8 jimaging-11-00043-f008:**
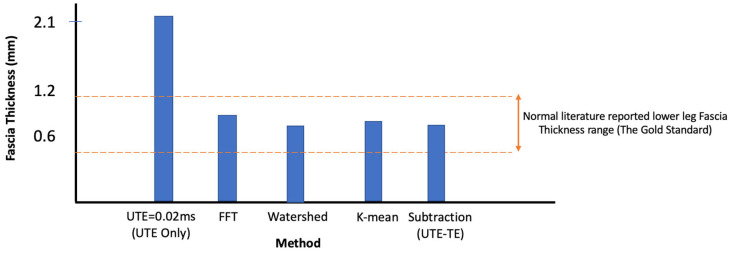
Comparison of deep fascia thickness measurements obtained using various image processing techniques with the normal range reported in the literature (gold standard). The UTE-only method shows higher thickness values, highlighting segmentation challenges without image enhancement, while other methods align closely with the normal range.

**Figure 9 jimaging-11-00043-f009:**
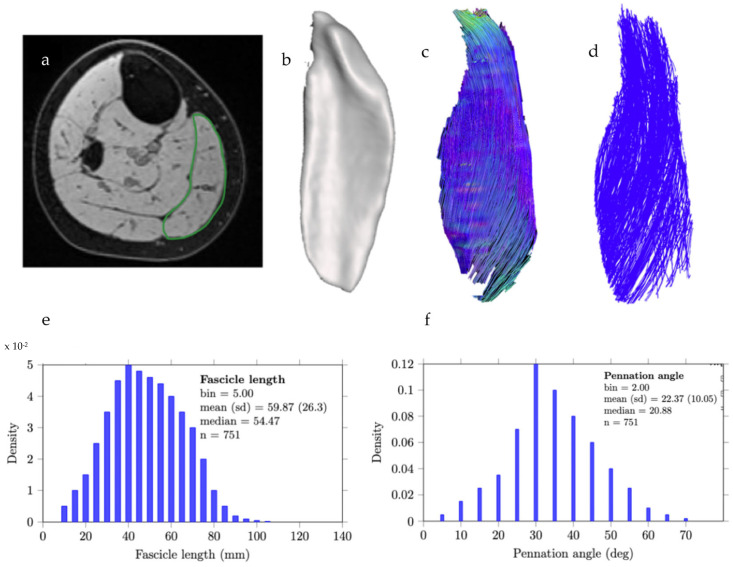
Overview of methods to measure muscle architecture from anatomical and DTI scans. (**a**) Transverse slice of a T1-weighted image approximately midway between the ankle and knee joint, showing outlines of the medial gastrocnemius (MG) muscles. (**b**) Anterior view of the three-dimensional reconstruction of the muscles. (**c**) Muscle DTI tractography reconstructions of the MG, where the tracts are colored according to their primary eigenvector direction. (**d**) Surface model of fascicle tract reconstructions of the medial gastrocnemius. (**e**,**f**) Muscle architecture parameters calculated based on DTI data. Architectural parameters computed are fascicle length (**e**) and pennation angles (**f**).

**Figure 10 jimaging-11-00043-f010:**
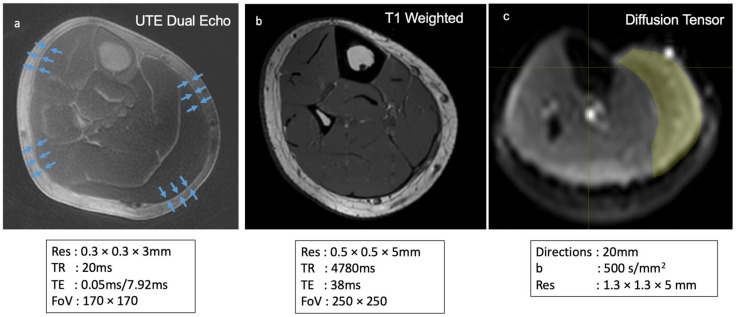
Summary of sequences used for imaging deep fascia, muscle fibers, and other connective tissue structures. Axial (**a**) dual-echo UTE subtraction imaging primarily highlights the deep fascia, providing detailed insights into its structure. Axial (**b**) T1-weighted imaging is used to visualize the anatomical structure of lower leg muscles, enabling accurate muscle volume calculations. Axial (**c**) DTI is employed to assess muscle fiber architecture, including measurements of fascicle length, pennation angle, and physiological cross-sectional area (PCSA).

## Data Availability

The datasets used and/or analyzed during the current study are available from the corresponding author upon reasonable request.

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
