# Peer review of "Imaging and Image Processing Techniques for High-Resolution Visualization of Connective Tissue with MRI: Application to Fascia, Aponeurosis, and Tendon"

_2313-433X, 2025, doi:10.3390/jimaging11020043_

Round 1

Reviewer 1 Report

Comments and Suggestions for Authors

The paper presents an approach to increase the visibility of thin connective tissues (e.g. fascia) within lower limb in MR T2 images.

First, the authors make an introduction to the problem of segmenting and visualizing thin connective tissue structures in MRI. This part is more less clear, however a more precise problem statement might appear. From various parts the reader finds that it is: optimizing the acquisition protocol parameters (sec. 1.1, 2.1, 2.4), deep fascia segmentation and measuring its thickness, muscle volume and cross section areas calculation (sec. 1.2), image contrast enhancement (sec. 5). However, the problem statement is mixed with brief descriptions of some image processing approaches and tools. The authors might provide e.g. a separate section with the main goals clearly listed. Then there is an introduction to some basic image processing and segmentation concepts (section 1.2). Discussing the pros and cons of manual, semi-automated and fully automated methods and just listing them makes almost no difference for the problem stated. Only some of the approaches listed have been utilized in the work, so why the others are mentioned? For example - why is ITK cited since there is no information that the authors have used this framework (apart from mentioning the ITKSnap software). Introduction to DTI in sec. 1.3 looks much better in the context of its usage in the presented method.

The methods section is the hardest to follow. While sections 2.1 and 2.4 are clear and obvious - just stating the acquisition parameters, the remaining ones seem to be used separately to get some (independent? Fig.7: are those 6 results somehow combined for the final segmentation or just shown for reference?) results. Flowcharts in Figs. 1 to 4 are so simple that may be replaced with bullet lists. On the other hand, the complete workflow of the study is hard to follow. A diagram showing which images (as there are mentioned 5 protocols in sec. 2.1) were used for which purpose would be helpful to the reader. Moreover, general methods' descriptions suit better to the introduction than to the methods section. In the methods section the reader would expect the detailed parameters of the methods used (e.g. kernel size, number of histogram bins, number of initial centroids and their values, etc.). The experiment cannot be repeated by others because the key details are not provided.

Results are presented only qualitatively and incomplete. Figs 5 and 6 present some of the acquired images, however how do those reflect the images acquired using any standard protocol? There is no comparison. The same concerns image processing results (Fig. 7) - there are no metrics nor comparison to any gold standard for the segmentation. Did a radiologist at least verify the the fascia regions shown in the image? At this stage the results may be interpreted as six individual ways used for edge detection and presuming those are the structure being searched for. Also from the method description it is not clear if the presented method provides the markers in Fig. 8, or those markers are added manually to show some properties of the resulting images. On a side note – why k-means clustering and watershed results are shown using a colormap? Wide range of colors – i.e. unique labels – indicate that an oversegmentation might have resulted. The outcome of those algorithms are just regions with unique values (labels) indicating the segmented regions. Without knowing the colormap used it does not show any usefulness - blue color appears also in other than fascia regions in the image. The only numerical values appear in Fig 9, however also without any validation if the results are correct and to what extent.

Finally, it is not clear if the authors developed and implemented their own software application or just used some existing tools and only used their capabilities.

The discussion section is lengthy and seems a repeat of the previous sections with some future plans. The reader would expect that the outcomes of the study are compared to other approaches showing that the presented method (presumably) is better than the already existing ones. Unfortunately, the paper does not provide it, therefore, leaving the reader unconvinced.

Language part of the paper is mostly correct and comprehensible. There are some minor editorial issues, especially missing or extra spaces and commas when citing references (e.g. line 33, 59 vs e.g. line 72, line 63 - missing space, line 188 - extra spaces, etc).

Reviewer 2 Report

Comments and Suggestions for Authors

Please see in pdf-file.

Reviewer 3 Report

Comments and Suggestions for Authors

This is a nice study exploring innovative imaging and processing techniques, focusing on ultrashort echo time (UTE) MRI, diffusion tensor imaging (DTI), and advanced image processing methods to achieve high-resolution visualization of connective tissues. The integration of these techniques presents significant advancements for musculoskeletal imaging, with potential applications in both clinical and research settings.

The manuscript is well-written, comprehensive, and demonstrates a clear understanding of the field. However, I have some questions of the authors:

1/ Could the authors provide further detail on the clinical applications and pathological tissues investigated in this study? Including examples of diseases or conditions, such as fascia-related injuries or aponeurosis tears, where these imaging advancements could be transformative, would enhance the discussion.

2/ Can the authors expand on how these imaging techniques can assist in the diagnosis or treatment of conditions involving connective tissues? Specific examples or case studies would be helpful for context.

3/ Could the authors ensure that the formatting of terms like Figure is consistent throughout the text, as it sometimes appears italicized and other times not?

4/ Please verify that all figures are appropriately referenced within the main text (e.g., Figure 4, Figure 8).

5/ To improve clarity, could the authors standardize terminology? For instance, ensure consistent use of terms like "fascia" and "deep fascia" throughout the manuscript.

6/ Some sections, particularly those discussing image processing algorithms, could benefit from condensation to improve readability and focus.

7/ Could the authors reorganize the Results section using clear and specific subheadings, such as "UTE MRI Results" and "DTI Tractography Findings"? This structure would make the findings more accessible to readers.

8/ Finally, could the authors expand on how this study compares to prior work in the field of connective tissue imaging? For example, a discussion of how the imaging resolution or segmentation accuracy achieved here compares with previous methods would provide valuable context and highlight the study's contributions.

Comments on the Quality of English Language

The Quality of English Language is excellent.

Round 2

Reviewer 1 Report

Comments and Suggestions for Authors

The Authors have considered previous remarks and the paper has been improved. Image processing details have been amended (maybe a bit too much as even the Matlab function names have been provided). There are, however, still some issues to be considered.

1. What segmentation has been performed using ITKSnap (l.70, l.88 and l. 281)? Its capabilities are limited mainly to snakes (active contours and the required prerequisites, that eventually could have been used as standalone methods, though). Was the fascia really segmented using ITKSnap as per l.88? If so - how? But from the remaining part of the manuscript it is clear that it was Matlab that was mainly used for image processing (and providing the outcomes in Fig.6). Maybe Snap was only used for resampling the MRI oblique slices into non-oblique axial plane instead of image segmentation? However, it is just reviewer's guess only.

2. How the "raw phase data" (l.154) was "processed to further increase the contrast..."? The paper presents that already reconstructed tomographic images have been used only. Maybe it is just a misunderstanding and sentence rephrasing would suffice.

3. Where was Canny edge detection used (l.76)? In the remaining part it is shown that edges were enhanced using FFT-based approach. What about other basic edge detection approaches (e.g. Laplace, Sobel)? Those have a control parameter related to spatial size that may be useful if the thickness is more-less known.

4. In what way does the K-Means clustering use pixel connectedness (l.198)? In l.217 the Authors state that "The kmeans function requires data to be in a one-dimensional format, so the original data was reshaped accordingly." The basic K-Means operates only on pixel intensities and does not use any form of spatial connectivity. Therefore, in the results there are disjoint regions with the same label, provided the original intensities fall to the same intensity cluster (range). Moreover, since the number of clusters was set to 3 (l.218), why the K Mean image (Fig.6f) is composed of only 2 colors (black and brown)? Maybe a better colormap could be used to present those 3 clusters?

5. In the watershed algorithm how it is possible to set the "desired number of clusters" (l.231)? The number of regions results from the level to which the ridges are "flooded". Maybe it is just a misunderstanding and sentence rephrasing would suffice.

6. Flowchart in Fig.2 shows that both the FFT and watershed are performed on the output image from the thresholding block. The same is stated in the caption of the figure. Is it really the case? Are those operations really performed on a binary image? From sec.2.2.4 it stems that binarization is performed (l.234) and the binary image is then used for distance map calculation prior to the watershed. But from sec.2.2.2 (l.178) it stems that the input to the FFT block is a grayscale image. The flowchart should present the input and output data paths correctly, not just block's place in the pipeline.

7. How the fascia thickness was measured? Was the human operator given the resulting images with segmented/enhanced contents and performed the measurements manually? As per Fig.6 the user is given 6 images. Or it was calculated as the average thickness of the region found during the (automated?) segmentation?

8. Since at least three different software packages have been used (ITKSnap, Matlab, MRtrix) it can be inferred that no dedicated CAD application has been developed. Therefore, using the presented approach seems complicated from the final user's point of view. Moreover, some values were fine-tuned (l.283) by trial and error (l.202, l.215), so how the method performs on “new” images that were not used during the design phase? Does the method make use of the whole 3D volume or performs the processing separately on each 2D slice? Considering high anisotropy of the voxel (ca 0.5x0.5x5mm) would there be a benefit of using the whole 3D volume in the processing? Could the Authors elaborate more on those three aspects (maybe in the limitations paragraph/section)?

9. Although some numerical results have been provided, the method still seems not fully validated. Do the Authors have a gold standard for the tissue segmentation? Then it would be possible to provide some measures indicating the quality of the segmentation (e.g. Dice Index, but there are many more). If there is no such database, it should be discussed in the limitations of the study. On the other hand, if the only way to validate e.g. the thickness is to ask radiologists to perform the measurements, one would expect at least two or three experts to perform the task (preferably - several times). Also experts might be presented with the enhanced as well as native images to verify if the presented approach is really helpful. Then some statistics could be calculated showing the usefulness of the method.

Structure of the manuscript still causes some confusion while reading.

10. Goals of the study are stated in two places: from l.96 and from l.125, consider making a single subsection or paragraph.

11. Usually, presenting first the general idea and the specific details later greatly eases problem understanding by others. The same concerns following the typical data flow: from image acquisition, through preprocessing, main processing and finally the results. Therefore, consider:
- moving sec.1.3 before 1.2 as it concerns imaging (acquisition) that is employed before the image processing,
- moving contents of sec.2.2.6 to sec.2.2, as it presents the general approach; subsequent subsections that provide the details of each step should also follow the order of performed operations. Please check the order of blocks in Fig.2 vs the order and contents of 2.2.x subsections.

12. There is still one main doubt regarding the deep fascia segmentation. From Fig.6 and the description it follows that there are presented 6 somehow separate methods/results:
- subtraction (sec.2.2.1, Fig.6a, not present in Fig.2),
- contrast enhancement (sec.2.2.5, Fig.6b, Fig.2 blocks 2 and 3, nb. note the different order of operations in the description and in the flowchart),
- Weiner filtered image after contrast enhancement (sec.2.2.5, Fig.6c),
- FFT edge detection (sec. 2.2.2, Fig.6d, nb. which of the preprocessed images is the input to this block? The Weiner filtered one?),
- K-Means clustering on IFFT image (sec. 2.2.3, Fig.6f),
- watershed on the preprocessed (Weiner filtered?) image (sec. 2.2.4, Fig.6e).
Finally, is the operator provided with those 6 images and is free to choose from them to do the measurements? Or the purpose of the figure is only illustrative?
Moreover, it seems that K-Means clustering on the IFFT image into 3 classes is just a nontrivial way of specifying edge values as "high", "medium" and "low". How do those correspond to the tissues in question?

13. Minor editorial:
l.28 and l.430 why the unit is mm^2?
almost throughout all the paper when citing there is missing space before the citation,
some paragraphs do not use justification in formatting (e.g. l.154 onward, l.198 onward, l.345 onward),
l.167 "flowchart" is a single word,
flowchart in Fig.1: the first block does "check the difference", however the result is indicated as a ratio; all the bullet lists could be moved inside the blocks in this flowchart,
l.177 last sentence is not necessary,
l.216, l.250 using first person in scientific writing is hardly used,
some lacking spaces (e.g. l.52 before hyphen, l.181),
extra spaces (e.g. l.230, l.248),
l.260 incorrectly referenced figure 8,
l.279 the section is so short that consider its merging with sec.2.2 Image Processing,
l.355 extra period before citations,
referencing subfigures in Fig.9 uses capital letters, remaining figures use lower case letters, consider using the same all the time; similarly, some figures use a hyphen in caption, some use a semicolon,
reference [17] seems corrupted, moreover its doi redirects to a publication by different authors, the remaining data (authors and journal) indicate a publication on Alzheimer's disease, which is irrelevant to the paper, consider rechecking all the references.

Round 3

Reviewer 1 Report

Comments and Suggestions for Authors

The Authors have responded to previous remarks and made the expected corrections. Now the paper can be followed much easier. In the reviewer's opinion the strengths of the paper are the MRI scanner setups, the weakest parts are the design of the image processing pipeline and its validation.

There are still minor issues to be resolved (mostly editorial) that Authors can resolve together with the Editorial team:
- Fig.1 is somewhat better, but still having two operations in a single block is misleading,
- Fig.2 "A n B" - shouldn't there be "A and B"?
- inconsistent writing, e.g. top hat vs Top Hat, k-means vs K Means vs kmeans,
- grammar, l.192, how the image can calculate something?
- change of style to very narrative, e.g. paragraphs starting in l. 192, 210, 249,
- sentence starting in l. 300 can be removed as the next paragraph starts with the same,
- still corrupted reference [17], some data missing in ref. [40] (journal?), [47] (title?), [63] (journal?),
- references [31] and [46] seem to be the same (despite some minor differences),
- missing or lacking spaces (l. 72, 131, 153, 286, 290, 311, 316, Fig.8 caption before the dash, 394, 395, 397, 566, 622, 623),
- empty lines (l. 151, 479),
- punctuation l. 438,
- formatting: text in italics l.290.
